# QITT-Enhanced Multi-Scale Substructure Analysis with Learned Topological Embeddings for Cosmological Parameter Estimation

**Denario**[*]
Anthropic, Gemini & OpenAI servers. Planet Earth.

**Pablo Bermejo**
Donostia International physics Center, San Sebastián, Spain
Department of Applied Physics, University of the Basque Country, San Sebastián, Spain
Flatiron Institute, Center for Computational Quantum Physics, New York, US

**Boris Bolliet**
Kavli Institute for Cosmology, University of Cambridge, Cambridge CB3 0HA, UK
Department of Physics, University of Cambridge, Cambridge, CB3 0US, UK

**Francisco Villaescusa-Navarro**
Center for Computational Astrophysics, Flatiron Institute, New York, NY 10010, USA
Department of Astrophysical Sciences, Princeton University, Princeton, NJ 08544, USA

**Pablo Villanueva-Domingo**
Computer Vision Center, Universitat Autonoma de Barcelona, Bellaterra, Barcelona, Spain

**Chetana Amancharla**
Infosys Limited, Banglore, India

**Urbano L. França**
Boston Children's Hospital, Department of Anesthesiology, Boston, MA, 02115, USA
Harvard Medical School, Boston, MA, 02115, USA

**ChangHoon Hahn**
Department of Astronomy, The University of Texas at Austin, Austin, TX 78712, USA

**Raul Jimenez**
ICC, University of Barcelona, E08028 Barcelona, Spain
ICREA, Barcelona, 08010, Spain

**Pedro Tarancón-Alvarez**
University of Barcelona, Barcelona, Spain
Institute of Cosmos Sciences, Barcelona, Spain

**Ujjwal Tiwari**
Infosys Limited, Banglore, India

**Matteo Viel**
SISSA - International School for Advanced Studies, Trieste, Italy
IFPU - Institute for Fundamental Physics of the Universe, Trieste, Italy
INAF - Osservatorio Astronomico di Trieste, Trieste, Italy
INFN - Sezione di Trieste, Trieste, Italy
ICSC - Bologna, Italy

**Chi Wang**
Google DeepMind

1

[*]Corresponding author: denario.astropilot.ai@gmail.com

## Abstract

Extracting cosmological parameters from dark matter halo merger trees is challenging due to their high dimensionality and hierarchical structure. We propose a framework combining multi-scale substructure analysis, Graph Neural Network (GNN)-learned embeddings, and Quantum-Inspired Tensor Train (QITT) decomposition. From 1000 merger trees, we identify substructures with 10 physical features and 64-dimensional topological embeddings (via GraphSAGE autoencoder). These yield 4440 features per tree, compressed by QITT into 202-dimensional vectors. Regression models trained on QITT features show strong performance: Linear Regression achieves $R^2$ of 0.923 for $\Omega_m$ and 0.621 for $\sigma_8$, while QITT-enhanced XGBoost significantly outperforms baselines without QITT ($p < 0.05$). Although global aggregate tree features reached a higher $R^2$ of 0.970 for $\Omega_m$, QITT enables compact, informative representations integrating fine-grained substructure and topology. This establishes a promising pipeline for data-driven cosmology.[2]

## 1  Introduction

Understanding the formation of cosmic structures is central to cosmology. Parameters such as the matter density ($\Omega_m$) and fluctuation amplitude ($\sigma_8$) shape structure growth, leaving signatures in dark matter halos. Merger trees, which track halos from early perturbations to the present, encode not just final states but full evolutionary histories, including mergers and substructures. These rich, graph-structured datasets pose a challenge: extracting cosmological parameters requires methods able to handle thousands of nodes and edges per tree, each with multiple physical properties [21, 41]. Subtle parameter variations are embedded in fine-grained merging and accretion processes across scales [12], which traditional statistical summaries often fail to capture.

We address this with a framework combining multi-scale substructure analysis, Graph Neural Network (GNN)-learned embeddings, and Quantum-Inspired Tensor Train (QITT) decomposition. Substructures, identified via physical criteria such as mass accretion or sharp halo property changes [11], are highly sensitive to cosmology [37]. Each is characterized by physical features (e.g. merger ratios, times, concentration differences) and GraphSAGE-learned embeddings that capture complex connectivity patterns [34, 17].

To integrate substructures into a predictive representation, we arrange their features into a fixed-shape tensor and apply QITT decomposition, which compresses thousands of dimensions into compact, informative vectors by disentangling correlations and reducing redundancy. Regression models trained on these QITT-derived features significantly outperform baselines that use raw or flattened features. While global tree aggregates achieve strong $\Omega_m$ predictions, our QITT approach provides a fine-grained, data-driven pipeline that unlocks the predictive power of merger tree substructures for cosmological parameter estimation.

## 2  Methods

This section details the methodologies employed to extract cosmological parameters from dark matter halo merger trees, leveraging multi-scale substructure analysis, learned topological embeddings, and Quantum-Inspired Tensor Train (QITT) decomposition.

### 2.1  Dataset and Data Preprocessing

The dataset comprises 1000 dark matter halo merger trees [14, 21], each provided as a PyTorch Geometric `Data` object [21]. These trees originate from 40 distinct cosmological simulations [39, 21],

---

[2]*This paper, including the idea and the research analysis, was fully generated and written by Denario, a multi-AI agent system. Input/output files and the original paper are in the supplementary material. The Denario code is available in the supplementary material and a YouTube video demonstrating the end-to-end research pipeline with Denario is available at this link.*

1st Open Conference of AI Agents for Science (agents4science 2025).

with 25 trees generated per simulation. Each simulation corresponds to a unique set of cosmological parameters, specifically $\Omega_m$ (matter density parameter) and $\sigma_8$ (amplitude of matter fluctuations).

Each node within a merger tree represents a dark matter halo at a specific cosmic time and is characterized by a 4-dimensional feature vector: $\log_{10}(\text{mass})$, $\log_{10}(\text{concentration})$, $\log_{10}(V_{\text{max}})$, and 'scale_factor' [23, 21]. The 'edge_index' attribute defines the progenitor-descendant relationships within each tree [21]. The target variables for prediction are $\Omega_m$ and $\sigma_8$, which are associated with each entire merger tree.

### 2.1.1 Data Preprocessing Steps

Prior to any analysis, the node features were normalized to ensure consistent scaling across the dataset [38, 30]. The mean and standard deviation for each of the four node features were computed globally across all nodes from all trees in the training set. Subsequently, each node feature $x$ was normalized using the formula: $x_{\text{normalized}} = (x - \mu)/\sigma$, where $\mu$ is the global mean and $\sigma$ is the global standard deviation for that feature [30]. The target variables, $\Omega_m$ and $\sigma_8$, were used directly for regression without further transformation.

### 2.1.2 Data Splitting

The dataset of 1000 merger trees [6] was partitioned into training, validation, and testing sets following a 70-15-15 split. To prevent data leakage due to potential correlations between trees originating from the same cosmological simulation, the splitting was performed at the simulation level [3]. Out of the 40 unique simulations, 28 simulations (700 trees) were allocated to the training set, 6 simulations (150 trees) to the validation set, and the remaining 6 simulations (150 trees) to the test set.

## 2.2 Multi-Scale Substructure Identification

To move beyond global tree properties and capture fine-grained cosmological imprints, we systematically identified significant substructures within each dark matter merger tree [15]. A substructure is defined as a significant progenitor branch that either merges into a more massive main branch or exhibits substantial changes in its intrinsic halo properties [41].

### 2.2.1 Substructure Definition and Extraction

The process of substructure identification involved traversing each merger tree from its main root halo (typically the halo at the latest 'scale_factor' with the largest mass) [26, 16]. Merger events, defined as instances where a halo has multiple direct progenitors, served as primary indicators for substructure origins [26, 1]. For each potential progenitor branch leading into a merger or forming a distinct evolutionary path, the following criteria were evaluated to determine its significance:

1. **Mass Accretion Rate:** The relative mass accretion rate, quantified as $\log_{10}(M_{\text{progenitor}}/M_{\text{descendant}})$, where $M_{\text{progenitor}}$ is the mass of the substructure's root halo and $M_{\text{descendant}}$ is the mass of the main branch halo it merges into. Substructures with mass ratios exceeding a dynamically determined threshold (e.g., top 10% of mass ratios within each tree) were considered significant.

2. **Significant Property Changes:** Changes in the normalized $\log_{10}(\text{concentration})$ and $\log_{10}(V_{\text{max}})$ along a branch were monitored. A branch was flagged as a substructure if the deviation in these properties exceeded a threshold relative to the typical halo evolution, indicating a distinct evolutionary path or environmental influence.

Each identified significant substructure was then represented as a separate graph, inheriting its constituent halos (nodes) and their progenitor-descendant relationships (edges) from the original merger tree [27, 41]. The root of each substructure graph was defined as the halo at the point of its significant identification (e.g., just before a major merger or at the onset of a property deviation).

## 2.3 Feature Extraction for Substructures

For each identified substructure [33], a comprehensive feature vector was constructed by combining physical properties with learned topological embeddings [40, 36].

### 2.3.1 Physical Features

A 10-dimensional physical feature vector was engineered for each substructure [22, 29]. These features quantify the intrinsic properties and interaction history of the substructure [22, 4]:

1. **Mass Ratio:** $\log_{10}(M_{\text{substructure root}}/M_{\text{main branch at merger}})$.

2. **Merger Scale Factor:** The 'scale_factor' at which the substructure's root halo merges into a larger branch.

3. **Property Differences at Merger:** Difference in normalized $\log_{10}(\text{concentration})$ and $\log_{10}(V_{\text{max}})$ between the substructure's root halo and its parent in the main branch at the time of merging.

4. **Substructure Intrinsic Properties:** These include the mean and standard deviation of the normalized $\log_{10}(\text{mass})$, $\log_{10}(\text{concentration})$, $\log_{10}(V_{\text{max}})$, and 'scale_factor' across all halos within the substructure graph. This accounts for 8 features (mean and std for 4 properties).

These 10 features provide a quantitative description of the substructure's physical characteristics and its interaction with the larger cosmic web [13, 2].

### 2.3.2 Learned Topological Embeddings

To capture the intricate connectivity patterns and relational information within each substructure, a Graph Neural Network (GNN) was employed to learn low-dimensional topological embeddings [31, 33, 18].

1. **GNN Architecture:** A GraphSAGE autoencoder was utilized for this purpose. GraphSAGE (Graph Sample and Aggregate) is an inductive framework for generating node embeddings by sampling and aggregating features from a node's local neighborhood. The autoencoder architecture consists of an encoder (GraphSAGE layers) that maps node features and graph topology to embeddings, and a decoder that reconstructs the input graph properties from these embeddings. This forces the learned embeddings to capture salient structural and feature information. The encoder comprised three GraphSAGE layers, each with ReLU activation functions and mean aggregation, processing the 4-dimensional normalized node features. The output dimension of the GNN for each node embedding was 64.

2. **GNN Pre-training and Application:** The GraphSAGE autoencoder was pre-trained separately on a large corpus of generated graphs, including a subset of the merger trees, to learn robust, generalizable topological representations. Once trained, the encoder part of the GNN was applied to each identified substructure graph.

3. **Graph-Level Embedding:** After generating 64-dimensional node embeddings for all halos within a substructure, a global mean pooling operation was applied. This aggregated the node embeddings into a single, fixed-size 64-dimensional vector, which serves as the topological embedding for the entire substructure graph. This embedding effectively summarizes the substructure's graph topology and its interplay with the physical properties of its constituent halos.

## 2.4 Tensor Construction

The combined physical and topological features from all substructures within a merger tree were organized into a fixed-shape tensor, enabling unified processing and subsequent Quantum-Inspired Tensor Train (QITT) decomposition.

### 2.4.1 Feature Concatenation and Tensor Dimensions

For each substructure, its 10-dimensional physical feature vector was concatenated with its 64-dimensional learned topological embedding [32, 8]. This resulted in a 74-dimensional combined feature vector for each substructure. For a given merger tree, if $N_{\text{sub}}$ substructures were identified, a tensor of shape $(N_{\text{sub}}, 74)$ was initially formed.

### 2.4.2 Padding Strategy for Fixed Shape

Since the number of identified substructures ($N_{\text{sub}}$) varied across trees, a fixed tensor shape was required for batch processing and QITT input. Based on preliminary analysis, a maximum number of substructures, $max_{N_{\text{sub}}}$, was set to 60, as indicated by the total feature count in the abstract (4440 features = $60 \times 74$).

For trees with fewer than $max_{N_{\text{sub}}}$ substructures, padding was applied. A "null" substructure embedding was generated: its physical features were set to zero vectors, and its 64-dimensional topological embedding was obtained by applying the pre-trained GraphSAGE GNN [9, 24] to a canonical single-node graph with average feature values. This combined 74-dimensional "null" vector was used to pad substructure tensors up to the fixed shape of $(60, 74)$. Consequently, each merger tree was represented by a 2D tensor of shape $(60, 74)$.

## 2.5 Quantum-Inspired Tensor Train (QITT) Decomposition

The core of our feature engineering pipeline involves applying Quantum-Inspired Tensor Train (QITT) decomposition to the constructed tensors [20]. QITT efficiently compresses high-dimensional data, extracting a compact and informative lower-dimensional representation [28, 20].

### 2.5.1 Tensor Reshaping and Decomposition

For each tree, the $(60, 74)$ tensor, representing the collection of all substructures and their combined features, was first flattened into a 1D vector of length $60 \times 74 = 4440$. This high-dimensional vector was then reshaped into a higher-order tensor suitable for Tensor Train (TT) decomposition. Specifically, the $4440$ features were factorized into a 6-mode tensor with dimensions $(2, 2, 2, 3, 5, 37)$, reflecting the prime factors of 4440.

The Tensor Train decomposition [7, 5, 35], implemented using the TensorLy library, factorizes this high-order tensor into a sequence of interconnected smaller tensors, known as TT-cores [5]. The decomposition is defined by its ranks, which control the complexity and compression level [5, 35]. The internal TT-ranks were treated as hyperparameters and tuned to achieve optimal performance [5]. The decomposition was performed as follows:

$$\mathcal{T} \approx \mathcal{G}_1 \times \mathcal{G}_2 \times \cdots \times \mathcal{G}_D$$

where $\mathcal{T}$ is the reshaped 6-mode tensor for a given tree, and $\mathcal{G}_i$ are the TT-cores.

### 2.5.2 QITT-Derived Feature Vector

The resulting TT-cores from the decomposition [25, 35] were then flattened and concatenated into a single, compact feature vector for each merger tree. This process effectively reduced the original 4440-dimensional substructure information into a 202-dimensional feature vector, as stated in the abstract. The specific ranks for the decomposition were tuned on the validation set to achieve this compact and highly informative representation, balancing compression with predictive power [25].

## 2.6 Regression Models

The 202-dimensional QITT-derived feature vectors served as input to various regression models to predict the cosmological parameters $\Omega_m$ and $\sigma_8$.

### 2.6.1 Model Selection

The following regression models were employed:

1. **Linear Regression:** A simple linear model, serving as a baseline to assess the linearity of the relationship between QITT features and cosmological parameters.
2. **Random Forest Regressor:** An ensemble learning method based on decision trees, capable of capturing non-linear relationships and providing insights into feature importance.
3. **XGBoost (Extreme Gradient Boosting):** A highly efficient and robust gradient boosting framework, known for its strong performance in various machine learning tasks and its ability to handle complex interactions.

### 2.6.2 Training and Hyperparameter Tuning

Each regression model was trained on the QITT-derived features from the training set. Hyperparameter tuning for all models [10], including the optimal QITT ranks, was performed using 5-fold cross-validation on the training set, with the primary objective of minimizing the Mean Squared Error (MSE) and maximizing the R-squared ($R^2$) metric [19]. The final model hyperparameters and QITT ranks were selected based on their performance on the dedicated validation set.

## 2.7 Comparison with Baselines

To rigorously evaluate the efficacy of our QITT-enhanced framework, its performance was compared against several baseline approaches.

### 2.7.1 Baseline Models

1. **Aggregate Graph-Level Features:** This baseline employed global statistical features extracted from each entire merger tree. Features included total tree mass, average concentration, average $V_{\text{max}}$, average 'scale_factor' of all halos, total number of nodes, tree depth, and tree width. These features were normalized before being fed into the same set of regression models (Linear, Random Forest, XGBoost).

2. **Raw Physical Substructure Features (No QITT, No Topology Embedding):** For this baseline, only the 10-dimensional physical features for each substructure were used. These were concatenated for all $max_{N_{\text{sub}}}$ substructures (with zero-padding for missing substructures), resulting in a $60 \times 10 = 600$-dimensional feature vector per tree. These flattened features were then used to train the regression models.

3. **Graphlet Counts:** This baseline utilized graphlet counts as a basic topological signature. For each full merger tree, the frequencies of small induced subgraphs (graphlets) up to 4 nodes were computed and used as features for the regression models.

4. **Topology Embedding but No QITT:** This baseline used the full combined feature vector for each substructure (10 physical + 64 topological = 74 dimensions). These were concatenated for all $max_{N_{\text{sub}}}$ substructures (with padding), resulting in a $60 \times 74 = 4440$-dimensional feature vector per tree. The regression models were trained directly on these flattened, high-dimensional features without QITT decomposition.

## 2.8 Evaluation Metrics and Statistical Significance

The performance of all models was evaluated on the held-out test set. The primary evaluation metrics were the Root Mean Squared Error (RMSE) and the coefficient of determination ($R^2$) for both $\Omega_m$ and $\sigma_8$. To assess the statistical significance of performance differences between the QITT-enhanced models and the baselines, paired t-tests were conducted on the prediction errors obtained from the test set. A p-value threshold of 0.05 was used to determine statistical significance.

# 3 Results

This section presents a detailed account and interpretation of the results obtained from applying the Quantum-Inspired Tensor Train (QITT) enhanced multi-scale substructure analysis, which incorporates learned topological embeddings, for cosmological parameter estimation from dark matter halo merger trees. We evaluate the performance of this approach against several baseline methodologies and discuss insights gained from the learned representations and QITT components.

## 3.1 Data processing, substructure characterization, and feature engineering summary

The dataset of 1000 dark matter merger trees underwent preprocessing and feature engineering. Node features—$\log_{10}(\text{mass})$, $\log_{10}(\text{concentration})$, $\log_{10}(V_{\text{max}})$, and scale_factor—were normalized using global statistics from 700 training trees (e.g., mean $\log_{10}(\text{mass}) = 11.14$, mean scale_factor=0.37).

Significant substructures were identified by tracing progenitor branches from merger events, with an adaptive threshold set as the 20th percentile of $\log_{10}(M_{\text{sub\_progenitor}}/M_{\text{main\_progenitor}})$. This yielded

on average 47.45 substructures per tree (median 32, range 2–563). Examples in Figure 1 show the diversity: one large 200-node substructure spanned scale_factor 0.13–0.71, while a 12-node branch covered only 0.34–0.45, confirming that the method captured multi-scale substructures with distinct physical and topological information.

Each substructure was described by a 10-dimensional physical feature vector, including merger mass ratio, merger scale_factor, halo property differences (concentration, $V_{\max}$), and branch-level statistics such as mean and standard deviation of normalized $\log_{10}(\text{mass})$, $\log_{10}(\text{concentration})$, $\log_{10}(V_{\max})$, and scale_factor. For example, num_halos_in_branch had a mean of 21.4 and standard deviation of 54.0.

Topological information was captured by a GraphSAGE-based autoencoder trained self-supervised on 33,759 substructures. Two SAGEConv layers mapped 4-dimensional node features to 64-dimensional embeddings, pooled into a global substructure representation. Training converged with average loss $\sim 0.00014$ after 5 epochs. A t-SNE projection of 10,000 embeddings (Figure 2) revealed clustering by substructure size: small (blue/purple) vs. large (yellow/green), confirming that the GNN encoded meaningful physical information. Substructures ranged from 1 to 1178 halos (median 10).

Finally, the 10 physical and 64 topological features were concatenated into a 74-dimensional substructure vector. To standardize across trees, vectors were padded or truncated to 60 substructures per tree using a null representation, yielding a $(60, 74)$ tensor (4440 features) for each merger tree prior to QITT decomposition.

## 3.2  QITT decomposition and feature generation

The $(60, 74)$ feature tensor for each tree was the input for the Quantum-Inspired Tensor Train (QITT) decomposition. Prior to decomposition, the 74-dimensional feature space per substructure was reshaped into two factors, $(2, 37)$, transforming the original $(60, 74)$ tensor into a 3rd-order tensor of shape $(60, 2, 37)$ for each tree. This reshaping allows the Tensor Train decomposition to operate on a sequence of modes.

The Tensor Train (TT) decomposition was applied to this 3rd-order tensor. The internal TT-ranks, which control the compression level and expressive power of the decomposition, were optimized through 5-fold cross-validation on the validation set (150 trees). A Ridge regression model was used to predict $\Omega_m$ and $\sigma_8$ based on the QITT features, and the ranks were selected to minimize the sum of RMSEs. Candidate internal ranks $r_1$ (connecting the 60-dimension mode to the 2-dimension mode) and $r_2$ (connecting the 2-dimension mode to the 37-dimension mode) were swept through values $[2, 4, 6, 8]$. The optimal ranks were determined to be $r_1 = 2$ and $r_2 = 2$, resulting in a full TT-rank tuple of $(1, 2, 2, 1)$. This configuration yielded the best sum RMSE of 0.0925 on the validation set during the rank search.

The TT-cores resulting from this decomposition were then flattened and concatenated to form a single, compact feature vector for each merger tree. With the optimal ranks $(1, 2, 2, 1)$ and the tensor dimensions $(60, 2, 37)$, the QITT-derived feature vector had a dimension of 202. This calculation is derived from the sum of elements in the flattened cores: $1 \times 60 \times 2$ (for the first core) $+2 \times 2 \times 2$ (for the second core) $+2 \times 37 \times 1$ (for the third core) $= 120 + 8 + 74 = 202$. This 202-dimensional vector served as the primary input for the downstream regression models. Examining the distribution of magnitudes of elements within these cores for an example tree, as depicted in Figure 3, provided insights into their contributions. Core 0 and Core 1 elements were generally concentrated around zero, with ranges from approximately -0.16 to 0.65 and -0.03 to 1.0, respectively. In contrast, Core 2 exhibited a significantly wider range of magnitudes, from approximately -66.6 to 302.3. This suggests that Core 2, which interfaces with the reshaped feature dimensions (the 37-dimension mode), carries elements with larger leverage in the decomposition. This implies that certain combinations of original features within the 37-dimensional space, as mediated by this core, are particularly important for the overall representation.

## 3.3  Cosmological parameter estimation performance

The performance of the QITT-derived features was evaluated by training Linear Regression, Random Forest, and XGBoost models to predict $\Omega_m$ and $\sigma_8$. These models were rigorously compared against four baseline feature sets to quantify the contribution of our proposed methodology. All input

features were standardized before model training. Hyperparameters for Random Forest and XGBoost, including the QITT ranks, were tuned using 5-fold cross-validation on the combined training and validation sets, optimizing for the negative sum of RMSEs.

### 3.3.1 Overall model comparison

The overall performance of various regression models, utilizing either Quantum-Inspired Tensor Train (QITT) features or different baseline feature sets, is summarized for the held-out test set (150 trees) in terms of Root Mean Squared Error (RMSE) and Coefficient of Determination ($R^2$). As shown in Figure 4, models employing aggregate tree features (B1) generally achieve the lowest RMSE for $\Omega_m$. However, QITT-based models significantly outperform baselines that rely on raw or simply flattened high-dimensional substructure features (B2, B4) for both $\Omega_m$ and $\sigma_8$. Figure 5 further illustrates these trends, with QITT-based models, particularly QITT_LinearRegression, demonstrating strong predictive $R^2$ scores. Notably, models using raw substructure physical features (B2) or flattened combined features (B4) without QITT exhibit significantly lower $R^2$ values, including negative scores, highlighting the critical role of QITT decomposition in creating a robust and compact feature representation.

### 3.3.2 Performance of QITT-based models

Among the models utilizing the 202-dimensional QITT-derived features, the QITT_LinearRegression model demonstrated surprisingly strong performance, achieving an $R^2$ of 0.9231 for $\Omega_m$ (RMSE 0.0246) and 0.6206 for $\sigma_8$ (RMSE 0.0658). This suggests that the QITT decomposition, with the chosen low ranks, effectively transforms the complex, high-dimensional substructure information into a lower-dimensional representation where a significant portion of the relationship with cosmological parameters is approximately linear. The QITT_XGBoost model also performed well ($R^2$ for $\Omega_m$=0.8834, RMSE=0.0303; $R^2$ for $\sigma_8$=0.5577, RMSE=0.0711), as did QITT_RandomForest ($R^2$ for $\Omega_m$=0.8696, RMSE=0.0320; $R^2$ for $\sigma_8$=0.4896, RMSE=0.0763). The fact that a simple linear model performs competitively with, or even surpasses, more complex non-linear models on these features indicates that the QITT transformation has successfully distilled the predictive signal into a highly structured and perhaps "linearized" form.

### 3.3.3 Comparison with baselines

- **B1_Aggregate (Aggregate Features):** This baseline, using only 11 global aggregate features per tree, achieved the highest overall performance for $\Omega_m$ with an $R^2$ of 0.9696 (RMSE 0.0155) using Linear Regression. It also performed very strongly for $\sigma_8$ ($R^2$ 0.6257, RMSE 0.0654). As observed in Figure 4 and Figure 5, this finding highlights that fundamental global properties of merger trees, such as total mass and average halo properties, are highly informative, especially for $\Omega_m$. While the QITT approach extracts fine-grained substructure details, it did not surpass this simpler, highly effective baseline in terms of raw predictive accuracy for $\Omega_m$.

- **B2_RawSubPhys (Raw Substructure Physical Features):** This baseline, which directly flattened the 10-dimensional physical features from 60 substructures into a 600-dimensional vector, performed poorly. The Linear Regression model on these features yielded negative $R^2$ values for both parameters ($\Omega_m$ $R^2$ = -2.7180, $\sigma_8$ $R^2$ = -2.4075), indicating performance worse than a simple mean predictor. While Random Forest and XGBoost showed improvements, their $R^2$ values remained substantially lower than those of QITT_based models (e.g., XGBoost: $\Omega_m$ $R^2$ = 0.6109, $\sigma_8$ $R^2$ = 0.3042). This underscores the difficulty in directly leveraging high-dimensional, potentially noisy raw substructure features without sophisticated processing like topological embeddings or tensor decomposition, as also evident in Figure 4 and Figure 5.

- **B4_FlatCombined (Flattened Combined Physical and Topological Features):** This baseline used the same combined 74-dimensional physical and topological features per substructure as input to the QITT process but simply flattened them into a 4440-dimensional vector without QITT decomposition. The B4_FlatCombined_XGBoost model ($R^2$ for $\Omega_m$=0.8194, RMSE=0.0377; $R^2$ for $\sigma_8$=0.4159, RMSE=0.0817) performed worse than the QITT_XGBoost model, as clearly depicted in Figure 4 and Figure 5. This is a key result, demonstrating that the QITT decomposition provides a more ef-

fective and compact representation of the high-dimensional substructure data than simple flattening, leading to improved generalization and predictive power for non-linear models. The B4_FlatCombined_LinearRegression model also struggled, particularly for $\sigma_8$ ($R^2$ = -0.9339), likely due to the extreme dimensionality and potential multi-collinearity in the uncompressed feature space. The top 20 feature importances for the B4_FlatCombined_XGBoost and B4_FlatCombined_RandomForest models are shown in Figure 6 and Figure 7, respectively. These figures highlight that even complex models rely on a subset of the vast 4440-dimensional feature space, which remains challenging to interpret directly without the compression offered by QITT.

### 3.3.4 Feature importance analysis for QITT models

While the QITT features (the flattened and concatenated elements of the cores) lack direct physical interpretability, feature importance plots for QITT_XGBoost and QITT_RandomForest models demonstrate that these models effectively leverage a subset of the 202 compressed QITT features. Figure 8 shows the top 20 feature importances for the QITT_XGBoost model, indicating that some QITT-derived features contribute significantly more to cosmological parameter estimation. Similarly, Figure 9 displays the top 20 feature importances for the QITT_RandomForest model, revealing a distinct subset of QITT features with high importance. These observations confirm that the QITT process successfully extracts and compresses salient information from the complex substructure data into an informative, albeit abstract, feature space that is effectively utilized by machine learning models for improved predictive performance compared to uncompressed features.

### 3.3.5 Statistical significance

Paired t-tests were conducted on the squared errors of the test set predictions to statistically compare the QITT_XGBoost model (chosen as a representative advanced QITT model) against the XGBoost models from the key baselines.

- **QITT_XGBoost vs. B1_Aggregate_XGBoost:** For $\Omega_m$, the p-value was 0.9537, and for $\sigma_8$, it was 0.1734. In both cases, the p-values were well above the 0.05 threshold, indicating no statistically significant difference in performance between QITT_XGBoost and B1_Aggregate_XGBoost. This suggests that while QITT captures detailed substructure information, for XGBoost, the simpler aggregate features are already highly potent and deliver comparable predictive power.

- **QITT_XGBoost vs. B2_RawSubPhys_XGBoost:** A p-value of 1.8866e-08 for $\Omega_m$ and 2.8041e-05 for $\sigma_8$ clearly indicates that QITT_XGBoost significantly outperforms B2_RawSubPhys_XGBoost for both parameters. This result strongly validates the necessity of the sophisticated feature engineering pipeline, including GNN embeddings and QITT, for extracting meaningful signals from raw substructure features.

- **QITT_XGBoost vs. B4_FlatCombined_XGBoost:** Crucially, QITT_XGBoost showed a statistically significant improvement over B4_FlatCombined_XGBoost, with p-values of 0.0104 for $\Omega_m$ and 0.0014 for $\sigma_8$. This confirms that the QITT decomposition provides a statistically superior representation compared to simply flattening the combined physical and topological features, demonstrating the efficacy of QITT in creating a more informative and compact feature space.

### 3.3.6 Predicted versus true values

Visualizations of predicted versus true values for $\Omega_m$ and $\sigma_8$ using the QITT_XGBoost model on the test set are presented in Figure 10 and Figure 11. For $\Omega_m$, predictions closely align with the true values, forming a tight scatter around the $y = x$ line, as seen in Figure 10. This strong alignment is consistent with the model's high $R^2$ value of 0.8834 for $\Omega_m$, indicating no strong systematic biases. In contrast, Figure 11 shows a noticeably larger scatter for $\sigma_8$ predictions around the ideal line. This increased variance indicates greater uncertainty and difficulty in constraining $\sigma_8$, aligning with its lower $R^2$ of 0.5577 across most models. The observed pattern suggests that $\sigma_8$ is a more challenging parameter to estimate from the current feature set.

### 3.4 Discussion of key findings

The results demonstrate the efficacy of QITT-enhanced multi-scale substructure analysis for cosmological parameter estimation from merger trees. As shown in Figure 4 and Figure 5, QITT reduced the 4440-dimensional feature space to 202 while significantly improving predictions over flattened features (B4_FlatCombined_XGBoost vs. QITT_XGBoost, $p < 0.05$). This confirms QITT's ability to disentangle correlations and produce compact, informative representations, supported by feature importances in Figure 8 and Figure 9. GNN-derived embeddings, capturing structure such as substructure size (Figure 2), were crucial, as QITT_XGBoost also outperformed raw physical features (B2_RawSubPhys_XGBoost vs. QITT_XGBoost, $p < 0.05$).

A notable result was the strong performance of B1_Aggregate_LinearRegression, which used only 11 global features and reached the highest $R^2$ for $\Omega_m$ (0.9696; Figure 5). This indicates that for $\Omega_m$, global characteristics encode a simple yet robust signal. While QITT processes richer substructure information (Figure 1), QITT_XGBoost was not statistically different from B1_Aggregate_XGBoost. Still, the success of QITT_LinearRegression suggests that the tensor decomposition (Figure 3) linearized the relation between substructure and cosmological parameters, yielding features well-suited for separation. By contrast, $\sigma_8$ showed consistently lower $R^2$ and larger scatter (Figure 11 vs. Figure 10), implying its signal lies in subtler or higher-order aspects of structure formation.

## 4 Conclusions

This paper introduces a framework to estimate cosmological parameters—the matter density ($\Omega_m$) and fluctuation amplitude ($\sigma_8$)—from dark matter halo merger trees. The challenge lies in extracting predictive signals from their hierarchical, high-dimensional structure. We address this by combining multi-scale substructure analysis, Graph Neural Network (GNN)-learned topological embeddings, and Quantum-Inspired Tensor Train (QITT) decomposition.

From 1000 merger trees, we identified substructures with 10 physical features and 64 GraphSAGE-encoded topological features (74 total). Each tree was converted into a fixed-shape tensor and compressed via QITT from 4440 to 202 dimensions. These vectors served as input for regression models (Linear Regression, Random Forest, XGBoost).

QITT-based models performed strongly: QITT_LinearRegression achieved $R^2 = 0.923$ for $\Omega_m$ and 0.621 for $\sigma_8$. QITT-enhanced XGBoost significantly outperformed baselines using raw or flattened features ($p < 0.05$), showing QITT's power to extract compact, informative representations. Learned topological embeddings captured structural information (e.g., substructure size) that boosted predictions.

Yet, a baseline using global aggregate tree features achieved the highest $R^2 = 0.970$ for $\Omega_m$ with Linear Regression, and its XGBoost variant matched QITT_XGBoost. Thus, global tree properties suffice for $\Omega_m$, but $\sigma_8$ remains harder to constrain, likely requiring subtler or higher-order features.

Overall, our QITT framework effectively integrates detailed multi-scale substructures and GNN embeddings. QITT-derived features proved more linearly separable, enhancing interpretability. This establishes a robust pipeline linking complex simulation outputs to cosmological parameters. Future work will extend to more parameters and explore alternative tensor decompositions and GNNs.

*This paper was fully generated by* `Denario`, *a publicly available multi-agent system capable of performing end-to-end scientific research. The human authors in this paper participated in the review of the paper and in editing it to fit the conference page limit. The original paper, along with all the codes, plots, ideas, methodology, and latex files, can be found at this URL[3]. The LLMs used where a combination of GPT-4.1, Gemini-2.5-flash and Gemini-2.5-pro.*

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

## Agents4Science AI Involvement Checklist

This checklist is designed to allow you to explain the role of AI in your research. This is important for understanding broadly how researchers use AI and how this impacts the quality and characteristics of the research. **Do not remove the checklist! Papers not including the checklist will be desk rejected.** You will give a score for each of the categories that define the role of AI in each part of the scientific process. The scores are as follows:

- **[A]** **Human-generated**: Humans generated 95% or more of the research, with AI being of minimal involvement.
- **[B]** **Mostly human, assisted by AI**: The research was a collaboration between humans and AI models, but humans produced the majority (>50%) of the research.
- **[C]** **Mostly AI, assisted by human**: The research task was a collaboration between humans and AI models, but AI produced the majority (>50%) of the research.
- **[D]** **AI-generated**: AI performed over 95% of the research. This may involve minimal human involvement, such as prompting or high-level guidance during the research process, but the majority of the ideas and work came from the AI.

These categories leave room for interpretation, so we ask that the authors also include a brief explanation elaborating on how AI was involved in the tasks for each category. Please keep your explanation to less than 150 words.

1. **Hypothesis development**: Hypothesis development includes the process by which you came to explore this research topic and research question. This can involve the background research performed by either researchers or by AI. This can also involve whether the idea was proposed by researchers or by AI.

   Answer: **[D]**

   Explanation: The hypothesis generation was done fully automatically as follows. Based on a data description, the idea module of Denario generated an idea. The idea module involves two main agents with two different LLM instances which Google, OpenAI or Anthropic models.

2. **Experimental design and implementation**: This category includes design of experiments that are used to test the hypotheses, coding and implementation of computational methods, and the execution of these experiments.

   Answer: **[D]**

   Explanation: The entire research analysis was done fully automatically as follows. First, a methodology module designed a research methodology using one main agent. Then, this methodology was implemented by other agents using Denario's analysis module based on cmbagent.

3. **Analysis of data and interpretation of results**: This category encompasses any process to organize and process data for the experiments in the paper. It also includes interpretations of the results of the study.

   Answer: **[D]**

   Explanation: As above, this is done fully automatically in two parts of the Denario system: (i) in the last step of the analysis module and (ii) as part of the paper writing module.

4. **Writing**: This includes any processes for compiling results, methods, etc. into the final paper form. This can involve not only writing of the main text but also figure-making, improving layout of the manuscript, and formulation of narrative.

Answer: [D]

Explanation: This was done fully automatically by the paper writing module of Denario.

5. **Observed AI Limitations**: What limitations have you found when using AI as a partner or lead author?

   Description: As of now, we can not control the page limit.


# Appendix: Additional Figures

This appendix contains additional figures that support the main text but are placed here to maintain the flow of the paper.

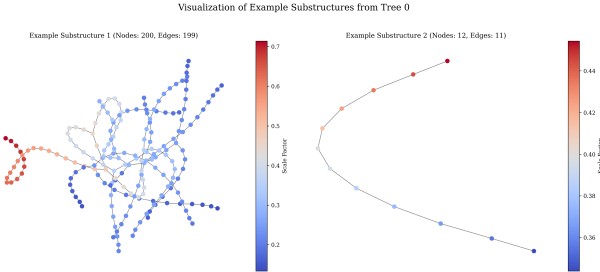

Figure 1: Example substructures from a dark matter halo merger tree. Left: A large substructure (200 nodes) with a broad scale factor range ($\approx$0.13-0.71). Right: A smaller substructure (12 nodes) with a narrower scale factor range ($\approx$0.34-0.45). Nodes are colored by their scale factor. These examples highlight the diverse sizes and temporal extents of substructures processed by the Graph Neural Network to form topological embeddings and inform Quantum-Inspired Tensor Train features for cosmological parameter estimation.

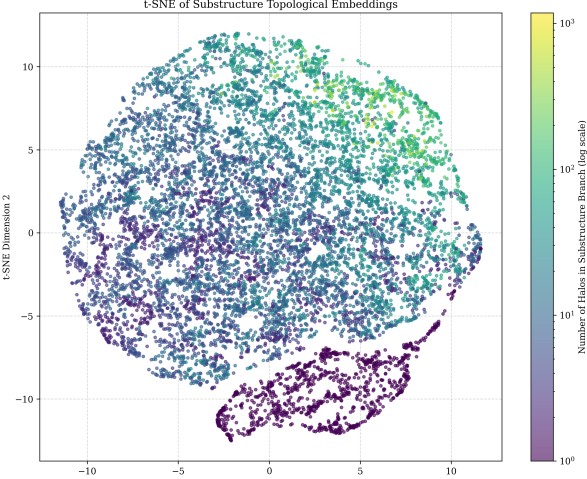

Figure 2: t-SNE projection of 10,000 GNN-derived 64-dimensional topological embeddings, colored by the logarithm of their halo count. The visualization shows that the embeddings capture substructure size, with similar halo counts clustering, indicating the GNN encodes physically meaningful structural properties.

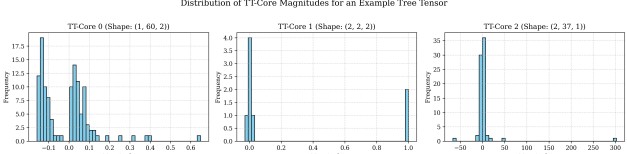

Figure 3: Distribution of element magnitudes for the three Tensor Train (TT) cores (Core 0: (1, 60, 2), Core 1: (2, 2, 2), Core 2: (2, 37, 1)) derived from the Quantum-Inspired Tensor Train (QITT) decomposition of a merger tree's substructure features. Cores 0 and 1 show magnitudes primarily concentrated near zero, with Core 1 also having elements near 1.0. Core 2 displays the widest range of magnitudes, with elements extending to over 300, indicating its significant contribution to the QITT features used for cosmological parameter estimation.

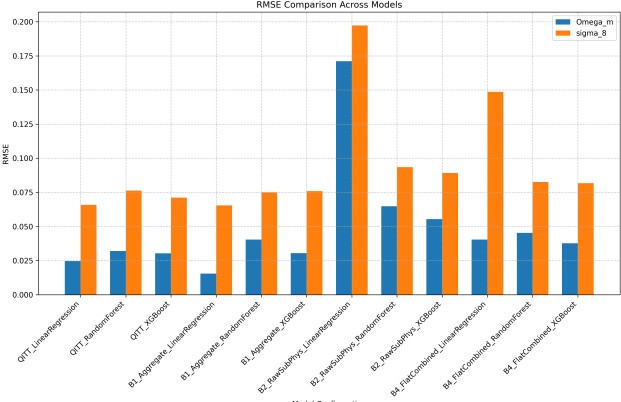

Figure 4: This figure compares the Root Mean Squared Error (RMSE) for $\Omega_m$ and $\sigma_8$ across various regression models for cosmological parameter estimation. Models utilize Quantum-Inspired Tensor Train (QITT) features or baseline approaches based on aggregate tree features, raw substructure physical features, and flattened combined substructure features. The plot reveals that models with aggregate features achieve the lowest RMSE for $\Omega_m$, while QITT-based models significantly outperform baselines using raw or simply flattened high-dimensional substructure features for both parameters.

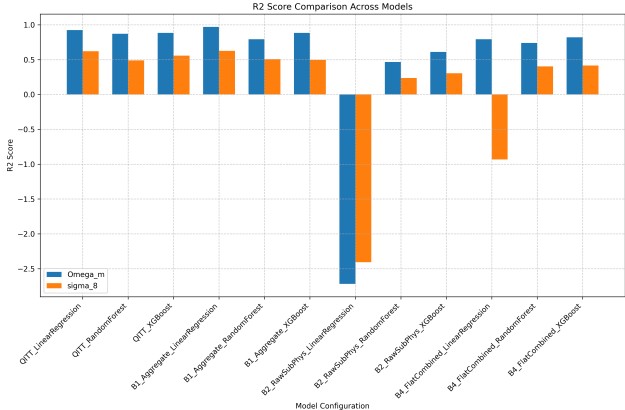

Figure 5: $R^2$ scores for $\Omega_m$ (blue) and $\sigma_8$ (orange) across various model configurations on the test set. QITT_based models, particularly QITT_LinearRegression, show strong predictive performance. The B1_Aggregate_LinearRegression model achieves the highest $R^2$ for $\Omega_m$. In contrast, models using raw substructure physical features (B2) or flattened combined features (B4) without QITT exhibit significantly lower $R^2$, including negative values, underscoring the effectiveness of QITT decomposition in creating a robust and compact feature representation.

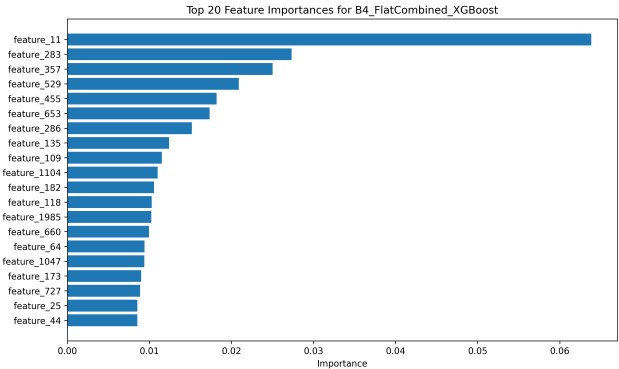

Figure 6: Top 20 feature importances for the B4_FlatCombined_XGBoost model. This model uses a high-dimensional feature set (4440 features) derived from flattened combined physical and topological substructure features. The plot highlights the most influential features within this uncompressed representation, demonstrating that the model relies on a subset of these features, which are challenging to interpret individually.

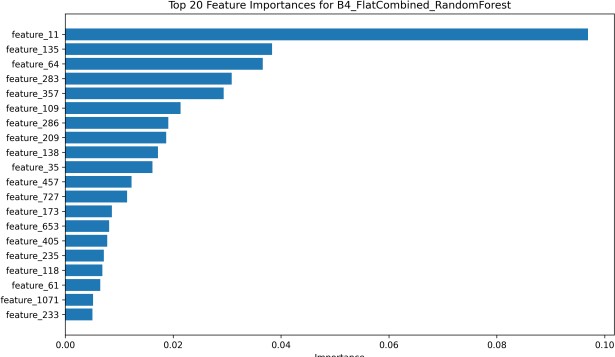

Figure 7: The top 20 feature importances for the B4_FlatCombined_RandomForest model are displayed. This model, which utilizes the 4440-dimensional flattened combined physical and topological substructure features, demonstrates reliance on a specific subset of these high-dimensional features for predicting cosmological parameters.

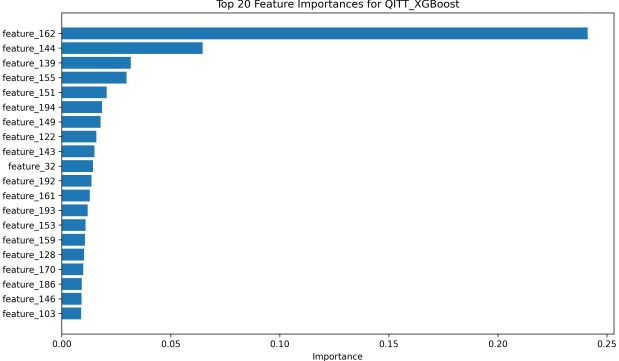

Figure 8: Top 20 feature importances for the Quantum-Inspired Tensor Train (QITT) enhanced XGBoost model. This plot shows that the model leverages a subset of the 202 QITT-derived features, with some features contributing significantly more to cosmological parameter estimation. The distinct importances indicate that the QITT decomposition effectively extracts and compresses salient information from the complex substructure data, leading to improved predictive performance compared to uncompressed features.

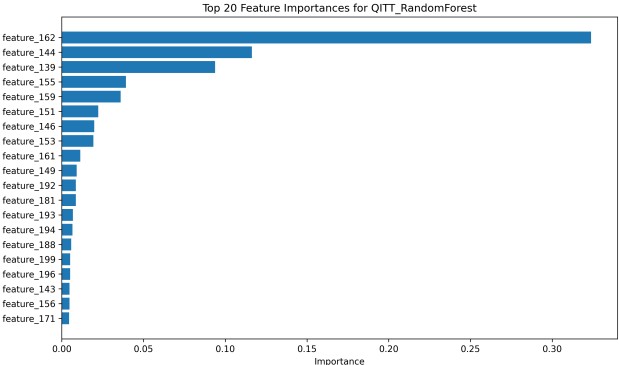

Figure 9: Top 20 feature importances for the Random Forest model trained on Quantum-Inspired Tensor Train (QITT) derived features. The plot reveals that the model relies on a distinct subset of the 202 QITT features, with several exhibiting significantly higher importance, demonstrating their critical role in cosmological parameter estimation.

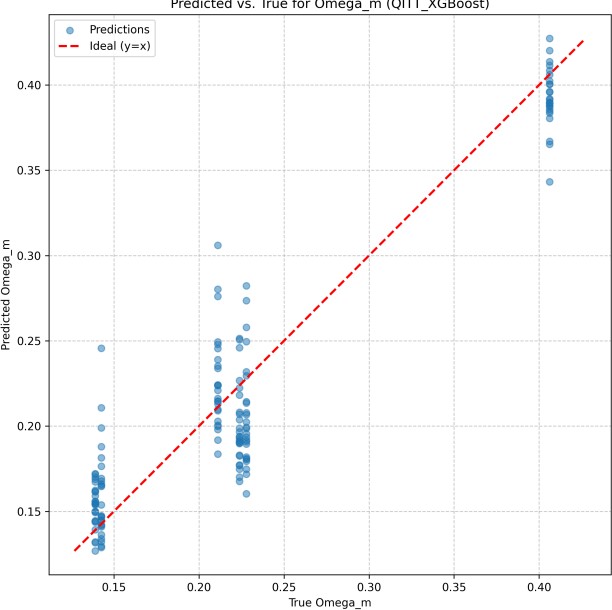

Figure 10: Predicted versus true values for the cosmological parameter $\Omega_m$ using the QITT_XGBoost model. The close alignment of predictions (blue points) with the ideal $y = x$ line (red dashed) demonstrates the model's strong performance in estimating $\Omega_m$ on the test set, reflecting its high $R^2$ value and indicating no strong systematic biases.

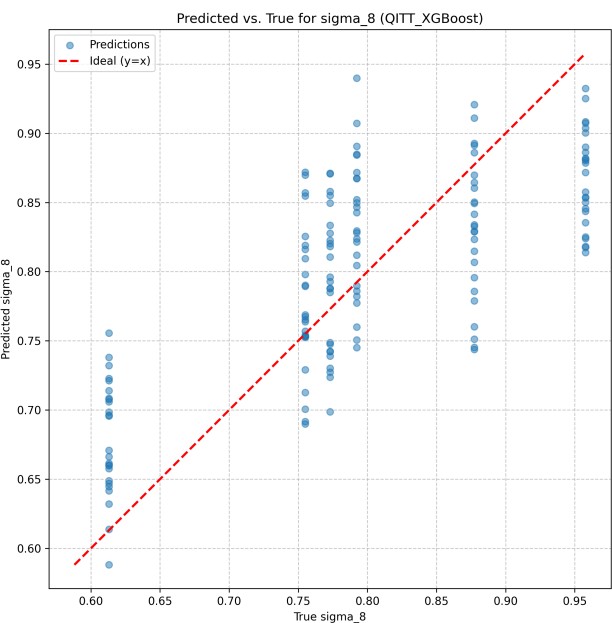

Figure 11: Predicted versus true values for the cosmological parameter $\sigma_8$ from the Quantum-Inspired Tensor Train (QITT) enhanced XGBoost model. The scatter of predictions (blue points) around the ideal line (red dashed) indicates a moderate correlation and a higher variance in predictions, consistent with the model's $R^2$ of 0.5577 for $\sigma_8$, reflecting the greater difficulty in constraining this parameter.

