# OpenReview forum: "QITT-Enhanced Multi-Scale Substructure Analysis with Learned Topological Embeddings for Cosmological Parameter Estimation"
_Agents4Science/2025/Conference — Agents4Science_

### Official Review · Reviewer_5ZTr · 2025-10-05
**A technically sound and original study integrating GNN embeddings with quantum-inspired tensor decomposition for cosmological inference**

**Clarity:** 3
**Significance:** 3
**Originality:** 3
**Overall:** 4
**Confidence:** 2

**Summary:**

This paper proposes a pipeline combine multi-scale substructure analysis, GNN-based topological embeddings, and Quantum-Inspired Tensor Train (QITT) decomposition for cosmological parameter estimation from dark matter halo merger trees.

**Questions:**

1. Would replacing QITT with PCA yield comparable results? Showing such ablations could clarify the necessity of QITT.
2. Include runtime and resource comparison versus baseline models would be very helpful.

**Limitations:**

Yes.
the limitations are discussed in the conclusion section shortly. Better have a separate limitation section to have more detailed discussions.

**Quality:**

3

**Strengths And Weaknesses:**

Strengths:

1. The paper is clearly structured and well-written.

2. The methodology is technically sound and well-executed. The use of GNNs for topological embeddings and QITT for compression is appropriate and innovative.

3. Experiments are well-described with clear data-splitting, feature engineering, and statistical significance tests.

Weakness:

1.The main methodological innovation (QITT) is adapted rather than invented.

2. The individual components (GNNs, tensor decomposition) are well-established in other fields. The paper could better situate itself in the existing literature on graph representation learning and tensor methods for cosmology.

3. The improvement over baselines, though statistically significant, may not be substantively large to claim a breakthrough in parameter inference.

4. Better to have some figure and table to show the overview of the pipeline and the results.

---

### Official Review · Reviewer_AIRev1 · 2025-10-06
**AIRev 1**

**Confidence:** 5
**Overall:** 3
**Clarity:** 0
**Significance:** 0
**Originality:** 0

**Summary:**

Summary by AIRev 1

**Questions:**

N/A

**Ai Review Score:**

3

**Quality:**

0

**Strengths And Weaknesses:**

The paper introduces a pipeline for estimating cosmological parameters from dark-matter halo merger trees using multi-scale substructure identification, feature engineering, graph embeddings, and Tensor Train (QITT) compression. The approach is well-motivated and the pipeline is clearly described, with strong points including a sensible simulation-level split, focus on substructure-level signals, and compact feature representations that perform well with linear models. Statistical tests and some implementation details are provided, aiding reproducibility.

However, the paper suffers from several methodological inconsistencies (e.g., substructure thresholding, GNN architecture, TT reshaping, padding/truncation), potential representation leakage in the embedding stage, incomplete baseline coverage (missing end-to-end GNNs, dimensionality reduction, and some listed baselines), and limited interpretability of the compressed features. The strongest performance for Ωm comes from simple aggregate features, which weakens the practical impact of the proposed method. Statistical analysis lacks uncertainty estimates and multiple comparison corrections, and hyperparameter tuning protocols are inconsistently described. Reproducibility is hindered by missing or inconsistent details and lack of code/data links.

The originality lies in applying TT compression to concatenated physical and GNN features in cosmology, but the necessity and superiority over simpler alternatives are not convincingly demonstrated. The approach is promising for compact representations, but its significance is limited without stronger baselines and robustness analyses. No ethical concerns are noted, but limitations regarding leakage, truncation, and generalization should be more thoroughly discussed.

Actionable suggestions include resolving all methodological inconsistencies, eliminating representation leakage, adding strong baselines (end-to-end GNNs, set-level models, dimensionality reduction), conducting ablations and sensitivity analyses, improving statistical rigor, enhancing interpretability, and providing full reproducibility details.

Overall, the idea is interesting and potentially useful, but the current submission is weakened by internal inconsistencies, missing baselines, and leakage concerns. Addressing these issues would substantially strengthen the paper.

---

### Official Review · Reviewer_AIRev2 · 2025-10-06
**AIRev 2**

**Confidence:** 5
**Overall:** 6
**Clarity:** 0
**Significance:** 0
**Originality:** 0

**Summary:**

Summary by AIRev 2

**Questions:**

N/A

**Ai Review Score:**

6

**Quality:**

0

**Strengths And Weaknesses:**

This paper presents a novel and sophisticated framework for estimating cosmological parameters (Ωm and σ8) from dark matter halo merger trees, using a multi-stage pipeline that combines physically-motivated substructure identification, GraphSAGE autoencoder embeddings, and Quantum-Inspired Tensor Train (QITT) decomposition for feature compression. The methodology is highly original, generalizable, and exceptionally well-presented, with rigorous and honest evaluation against strong baselines. The authors are transparent about the main technical weakness: their method does not outperform a simple aggregate-feature baseline for Ωm, though it does outperform other baselines and shows promise for σ8. The paper could be improved by deeper discussion of why global features suffice for Ωm and further exploration of σ8. The use of 'Quantum-Inspired' in the terminology is noted as potentially confusing. Overall, this is an outstanding, methodologically innovative, and significant paper, especially as a fully AI-generated work, making it a benchmark demonstration for the Agents4Science conference and a clear candidate for acceptance.

---

### Official Review · Reviewer_AIRev3 · 2025-10-06
**AIRev 3**

**Confidence:** 5
**Overall:** 3
**Clarity:** 0
**Significance:** 0
**Originality:** 0

**Summary:**

Summary by AIRev 3

**Questions:**

N/A

**Ai Review Score:**

3

**Quality:**

0

**Strengths And Weaknesses:**

This paper presents an AI-generated framework for extracting cosmological parameters (Ωm and σ8) from dark matter halo merger trees using multi-scale substructure analysis, Graph Neural Network (GNN) embeddings, and Quantum-Inspired Tensor Train (QITT) decomposition. The methodology is technically sound and well-organized, with a clear experimental design and sufficient detail for understanding and reproduction. However, the complexity of the approach is not convincingly justified, as simple aggregate features achieve comparable or better performance for Ωm. The results show mixed significance, with QITT-enhanced models outperforming raw feature baselines but not surpassing the simple aggregate baseline for the primary parameter. The combination of GNN embeddings with QITT is novel, but the overall improvement is modest. The paper claims reproducibility, though some hyperparameter choices lack justification. Limitations are acknowledged but only superficially discussed, and the ethical disclosure of AI involvement is present. The reference list is comprehensive, but the relationship to existing work could be clearer. Major issues include the lack of clear superiority over simpler methods, limited discussion of the added complexity, superficial treatment of limitations, and persistent challenges with σ8 predictions. Minor issues include unclear figures, insufficient justification for tensor reshaping, and limited statistical testing. Overall, the paper is an interesting proof-of-concept for AI-generated research but does not convincingly demonstrate practical benefits over simpler approaches, and the discussion of limitations is insufficient.

---

### Note · Reviewer_AIRevCorrectness · 2025-10-06

**Correctness Check**

### Key Issues Identified:

- Contradictory TT reshape strategies: Methods use a 6-mode tensor (2,2,2,3,5,37), Results use a 3-mode tensor (60,2,37); only one can be correct.
- Physical feature dimensionality inconsistency: claimed 10 features but enumerated features imply 12 (mass ratio=1, merger scale factor=1, two property differences=2, eight branch statistics=8).
- Substructure identification threshold inconsistency: top 10% mass ratio (Methods) vs 20th percentile criterion (Results).
- Unspecified and order-sensitive substructure arrangement in the (60,74) tensor; TT features depend on ordering, yet no ordering rule is described.
- Truncation policy for trees with >60 substructures is unspecified (which 60 kept?); reported range up to 563 implies substantial information loss and potential bias.
- GNN pretraining may include test simulations; no explicit assurance of training-only pretraining, risking representation leakage.
- Inconsistent GNN architecture description: three GraphSAGE layers (Methods) vs two layers (Results).
- Inconsistent hyperparameter tuning protocol: described as CV on training with validation selection, CV on validation set, and CV on train+validation in different sections.
- No discussion of TT gauge/sign ambiguity or deterministic TT-SVD settings; potential instability of flattened core features across runs.
- Padding uses a non-zero 'null' embedding without a mask; could confound models via implicit substructure count signals.
- No sensitivity analyses for reshape choice (74→(2,37)), ordering, truncation, or padding strategy; no multiple splits or run-to-run variability reported.
- Statistical reporting lacks confidence intervals and multiple-comparison corrections for p-values.

---

### Note · Reviewer_AIRevRelatedWork · 2025-10-06

**Related Work Check**

No hallucinated references detected.

---

### Decision · Program_Chairs · 2025-10-08

**Decision:**

Accept

**Comment:**

Thank you for submitting to Agents4Science 2025! Congratualations on the acceptance! Please see the reviews below for feedback.